# Fentanyl Induces Novel Conditioned Place Preference in Adult Zebrafish, Disrupts Neurotransmitter Homeostasis, and Triggers Behavioral Changes

**DOI:** 10.3390/ijerph192013533

**Published:** 2022-10-19

**Authors:** Yuanzhao Wu, Anli Wang, Lixiang Fu, Meng Liu, Kang Li, Song Chian, Weixuan Yao, Binjie Wang, Jiye Wang

**Affiliations:** 1Key Laboratory of Drug Prevention and Control Technology of Zhejiang Province, The Department of Criminal Science and Technology, Zhejiang Police College, Hangzhou 310053, China; 2National Engineering Laboratory of Intelligent Food Technology and Equipment, Zhejiang Key Laboratory for Agro-Food Processing, Fuli Institute of Food Science, College of Biosystems Engineering and Food Science, Zhejiang University, Hangzhou 310058, China; 3Office of Criminal Science and Technology, Xiaoshan District Branch of Hangzhou Public Security Bureau, Hangzhou 310002, China

**Keywords:** zebrafish, fentanyl, shoaling, conditioned place preference, neurotransmitter

## Abstract

Abuse of new psychoactive substances increases risk of addiction, which can lead to serious brain disorders. Fentanyl is a synthetic opioid commonly used in clinical practice, and behavioral changes resulting from fentanyl addiction have rarely been studied with zebrafish models. In this study, we evaluated the rewarding effects of intraperitoneal injections of fentanyl at concentrations of 10, 100, and 1000 mg/L on the group shoaling behavior in adult zebrafish. Additional behavioral tests on individual zebrafish, including novel tank, novel object exploration, mirror attack, social preference, and T-maze memory, were utilized to evaluate fentanyl-induced neuro-behavioral toxicity. The high doses of 1000 mg/L fentanyl produced significant reward effects in zebrafish and altered the neuro-behavioral profiles: reduced cohesion in shoaling behavior, decreased anxiety levels, reduced exploratory behavior, increased aggression behavior, affected social preference, and suppressed memory in an appetitive associative learning task. Behavioral changes in zebrafish were shown to be associated with altered neurotransmitters, such as elevated glutamine (Gln), gamma-aminobutyric acid (GABA), dopamine hydrochloride (DA), and 5-hydroxytryptamine (5-HT). This study identified potential fentanyl-induced neurotoxicity through multiple neurobehavioral assessments, which provided a method for assessing risk of addiction to new psychoactive substances.

## 1. Introduction

Overdose on opioids can cause paranoia, anxiety, panic, and delusions of victimization in humans [1], and may even lead to violent crime [2]. Fentanyl and its derivatives are synthetic opioids with strong analgesic and narcotic effects. Although clinical use of fentanyl has been strictly regulated, there have been cases of drugs or narcotics adulterated with fentanyl to enhance sedation, resulting in acute poisoning and even death [3]. More than half of all drug overdose deaths in the United States in 2020 were caused by fentanyl and its variants, which resulted in the deaths of 48,000 people that year [4]. In the studies of fentanyl-related deaths, the average blood concentration of fentanyl was 24 μg/L [5] and 57.9 μg/L [6]. However, the neuro-behavioral damage caused by addictive doses of fentanyl has not been well understood. Studies on the neuro-behavioral toxicology induced by fentanyl may contribute to development of effective treatments for addiction.

Fentanyl has strong sedative effects and a high risk of addiction due to its highly esterophilic nature, which allows it to rapidly cross the blood–brain barrier and then bind potently to opioid receptors in the brain [7]. Short-term overdose or long-term clinical use of fentanyl could lead to brain disorders [8] accompanied by abnormal social behaviors, such as disorientation, attention deficit, and memory impairment [9,10]. In addition to neurotoxicity, studies have indicated that clinical overdoses of fentanyl could induce chest wall rigidity, which, in turn, leads to respiratory depression and increased risk of death. These effects may be related to norepinephrine and cholinergic pathways [11]. Fentanyl has demonstrated similar toxicity in rodent and primate models, including risk of addiction [12,13] and respiratory depression [14]. In mice, fentanyl has also been shown to induce bradycardia [15] and nosocomial sensitization [16]. New animal models, such as fish models with rich behavioral parameters, could allow for more comprehensive behavioral toxicity evaluation of fentanyl.

In recent years, several protein receptors for opioids, cannabinoids, nicotine, and ethanol have been identified in zebrafish, promoting zebrafish as a useful model for drug addiction research [17]. Compared to rodent models, zebrafish models require less space for behavioral testing, less expenditure on experimental individuals, and less requirements for animal welfare. It was found that the fentanyl metabolic pathway in zebrafish was similar to that of rats, implying that toxicity results from zebrafish could be verified by other models [18]. In addition, zebrafish larvae have been used to validate cardiotoxicity due to fentanyl exposure [19] and respiratory depression toxicity [20]. Studies showed that exposure to fentanyl impaired the ability of zebrafish larvae to respond to light and dark stimuli [21]. However, the complex behavioral changes induced by fentanyl exposure have not been further revealed by zebrafish larvae, possibly due to the fact that larval brain function has not yet fully developed.

Adult zebrafish display complex behaviors, including shoaling, anxiety, aggression, socialization, exploration, and memory, making them an ideal model for studying the behavioral toxicity of drugs [22]. Conditioned place preference (CPP) is an important experimental method for evaluating drug addiction based on the principles of Pavlovian conditioning, which could reflect the rewarding nature of drugs by analyzing the association between drugs and environmental stimuli established in model animals [23]. According to previous research, CPP could be achieved in zebrafish using opioids, ethanol, and nicotine by immersing the subjects in the respective drug solutions [17]. Fentanyl at concentrations of 0.004–0.16 mg/L has also been shown to induce CPP in adult zebrafish by drug immersion [24]. However, the individual zebrafish behavioral differences in these experiments could significantly affect evaluation of reward effects. In contrast, group shoaling of zebrafish exhibited unique behavioral patterns based on social stability and group memory [25,26]. With the advancement in monitoring equipment and image tracking algorithms, it is now feasible to quantify group behavior in zebrafish, including regional preferences and social habits [27]. Individual zebrafish may also be compelled to certain locations in the shoal by the more vehement preferences of their conspecifics, thus avoiding individual-induced instability [28]. The shoaling behavior of zebrafish has been used in CPP tests to investigate drug reward effects [29,30,31].

In this study, we expected that high fentanyl concentrations would induce conditioned place preferences of group shoaling in adult zebrafish. Considering the variability in drug absorption by individual zebrafish, we used intraperitoneal injection, which could provide more accurate dose conversion data. Using different behavioral models, we evaluated the effects of fentanyl on anxiety, aggression, social preference, and memory in zebrafish. In addition, we investigated the neurotransmitter alterations in L-glutamine (Gln), γ-aminobutyric acid (GABA), dopamine hydrochloride (DA), and 5-hydroxytryptamine (5-HT) and the hormone change in cortisol in zebrafish brain after fentanyl treatment. To our knowledge, this was the first time a conditioned place preference model has been used to study the rewarding effects of fentanyl on zebrafish shoaling behavior. These findings could provide a method for future research on addiction-induced behavioral toxicity of other new psychoactive substances.

## 2. Materials and Methods

### 2.1. Zebrafish Maintenance

In this study, adult wild-type zebrafish (AB strain) were obtained from the Wuhan Zebrafish Center (Wuhan, China) and kept in a commercial recirculating tank system (Shanghai Haisheng Biotech, Shanghai, China). Adult zebrafish lived in a steady aquatic environment with a temperature range of 27.5–28.5 °C. The water in the aquarium system was constantly filtered and oxygenated (pH 7.0–7.5, conductivity at 500–600 μS). Newly hatched brine shrimp were used for feeding twice daily. A 14-h/10-h light/dark cycle was used to simulate the environment’s illumination (light started at 6:00 and light ended at 20:00 each day). The Experimental Animal Ethics Committee ZJU-IACUC of Zhejiang University authorized the zebrafish experimentation techniques (ZJU20220147).

### 2.2. Chemicals

Fentanyl hydrochloride (CAS number: 1443-54-5, >99% purity) was acquired from Shanghai Yuansi Standard Science and Technology Co., Ltd. (Shanghai, China). Thermo LabTower EDI 15 Integrated ultrapure water system was used to generate deionized water to prepare fentanyl stock solutions. Methanol and acetonitrile were of chromatographically pure quality (Merck, Darmstadt, Germany). Gln, GABA, DA, 5-HT, cortisol, and formic acid were bought from Aladdin Industrial Corporation (Shanghai, China).

### 2.3. Zebrafish Intraperitoneal Injection Protocol

Zebrafish used for the study were 6–8 months old and weighed 0.5 ± 0.1 g. Prior to the start of the experiment, the zebrafish were maintained in a 28 °C room without food for two days. During this period, baseline measurements of fish place preference were taken once a day for 20 min in the test tank to ensure that the fish did not show a significant preference for white or black spotted areas, i.e., to ensure that the dotted pattern was a neutral stimulus for the fish. On the third day, the zebrafish were placed in a CPP test tank for 20 min of training after intraperitoneal injection, a procedure that was repeated once a day for a total of four days. Specifically, zebrafish were immersed in cold water at a temperature of 12 °C for hypothermia prior to intraperitoneal injection, which was reported to be a better option than tricaine methanesulfonate for anesthesia [32]. The fish immediately started to be injected after they stopped moving or only the gills were moving. With an injection volume of 5 μL, a Hamilton syringe (0.1 mL; Hamilton) carrying the solution on a 30-gauge needle was gently injected into the midline of the abdomen behind the pectoral fins. On days 3 and 5, zebrafish in the experimental groups were injected with different concentrations of fentanyl solution and then placed through a spacer in the white area with black spots. On days 4 and 6, zebrafish in all groups were injected with saline and then placed through a spacer in the white area without black spots. The zebrafish in control group were always given a 5 μL saline injection. Fentanyl concentrations of 10 mg/L, 100 mg/L, and 1000 mg/L were chosen for trials after preliminary tests, corresponding to body weight dosages of 0.1 mg/kg, 1 mg/kg, and 10 mg/kg, respectively. The experiment was repeated three times, with each concentration group including 16 fish.

### 2.4. Behavioral Test

At the end of the intraperitoneal injection experiment, we performed fish shoaling behavior assessments and conditioned place preference trials in order to examine the rewarding effects of fentanyl. Afterwards, we examined the performance of individual zebrafish in novel tank, novel object exploration, mirror attack, social preference, and T-maze memory (Figure 1). Zebrafish behaviors were collected using an infrared camera from Zebracube (Viewpoint), and the data were analyzed by Zebralab (Viewpoint) and DanioVision (Noldus) software. Behavioral surveys were conducted from 09:00 to 15:00 daily, with different types of behavioral tests performed at intervals of 4 h or more.

#### 2.4.1. Conditional Place Preference Test

The conditional place preference test in zebrafish is a useful method to evaluate the rewarding effects of drugs [24]. A rectangular test tank (30 × 20 × 15 cm, 8 L of water) was divided into two equally spaced zones, with four 3-cm-diameter black spots evenly distributed in one zone as the fentanyl pairing zone (Figure 2A). In the Zebracube, a top-mounted infrared camera was used to record the behavior of zebrafish group for 20 min under a light intensity of 100 lux. The percentage of the average number of zebrafish in the fentanyl-paired and non-drug-paired zones was calculated at 2-min intervals. Sixteen zebrafish per concentration were selected for behavioral test, with 16 fish placed in each tank.

#### 2.4.2. Shoaling Test

Shortly after hatching, shoaling behavior begins in zebrafish, and cohesiveness in shoaling behavior is a frequent way of evaluating the impact of toxicants on the social behavior of zebrafish [33]. Several measures of shoaling are quantified, including the nearest neighbor distance (NND: the distance between each individual and its closest neighbor), inter-individual distance (IID: the mean distance from each individual to all the other fish), and swimming speed (the mean of the momentary speeds of all the fish). The zebrafish group shoaling behavior was analyzed using a rectangle test tank (30 × 20 × 15 cm, 8 L of water). Sixteen zebrafish per concentration were selected for shoaling test, with 16 fish placed in each tank. The behavior of each zebrafish group was recorded by an infrared camera on top of the Zebracube tank for 20 min with light intensity of 100 lux. At 2-min intervals, NND and IID determined from any two zebrafish and the average swimming speed of all the fish were monitored.

#### 2.4.3. Novel Tank Test

As described in previous studies [34], we used a new tank test to assess the anxiety-like behavior of zebrafish. The paradigm is a well-established, standardized assay to assess the natural response of zebrafish in new environments [33,35]. Sixteen zebrafish from each concentration were selected for the novel tank trial, and a total of four tanks were used simultaneously, with one fish placed in each tank. The trapezoidal tank used for the new tank test was 15.0 cm long, 28.0 cm wide at the top, 23.0 cm wide at the bottom, and 6.0 cm high. Without an acclimation period, the locomotor behavior of each zebrafish in the new tank was recorded individually for 3 min. The tank is divided into two equal horizontal areas, the top and the bottom, by drawing a demarcation line on the outside of the tank. The total distance travelled by the zebrafish in the tank, the time spent in the top area, the latency to enter the top area, and the number of transitions into the top area were analyzed.

#### 2.4.4. Novel Object Exploration Test

Sixteen zebrafish from each concentration were selected for novel object exploration test in a rectangular tank (20 × 10 × 10 cm, 2 L of water). A total of four tanks were used simultaneously, with one fish placed in each tank. The tank was divided into two equal zones (zone of new object and zone without new object), and a green cylinder with a diameter of 4 cm was placed in the center of one of the zones (zone of new object). The behavioral data of each zebrafish were recorded for 5 min through an infrared camera on top of the tank. The first 2 min were used for adaptation and the second 3 min were used for behavioral data analysis, including the total distance travelled by the zebrafish in the tank, the time spent in the new object area, the latency to enter the new object area, and the number of transitions into the new object area.

#### 2.4.5. Mirror Attack Test

Aggression in zebrafish was defined as biting, short intervals of intermittent swimming, or rapid swimming towards the tank wall close to the mirror image, aimed at attacking virtual conspecifics [36]. Sixteen zebrafish from each concentration were selected for mirror attack test using a rectangular water tank (20 × 10 × 10 cm, 2 L of water) [37]. A total of four tanks were used simultaneously, with one fish placed in each tank. A mirror was attached to one of the tank’s sides. The mirror area was defined as a distance of 5 cm near to the mirror. The total distance travelled by the zebrafish in the tank, the time spent in the mirror area, the latency to enter the mirror area, and the duration of fast swimming (velocity greater than 60% of the mean velocity was defined as fast movement) in the mirror area were analyzed for a period of 5 min.

#### 2.4.6. Social Preference Test

Sixteen zebrafish from each concentration were selected for social preference test using a rectangular tank (30 × 10 × 10 cm, 3 L of water) [34]. A total of four tanks were used simultaneously, with one fish placed in each tank. The tank was divided into three rooms of equal size, two of which were joined to form a large chamber (the big chamber) and separated by a transparent acrylic panel from the third chamber (the small chamber). The tested zebrafish were permitted unrestricted movement in the main chamber, with five randomly selected adult zebrafish from the recirculating tank system being placed in the small chamber. Five minutes of infrared camera observation from the top of the tank was used to record behavioral data for each zebrafish subjected to testing. The first two minutes of data were analyzed for adaptation, while the last three minutes were behavioral analysis, including the total distance travelled by the zebrafish in the tank, the time spent in the social area, the latency to enter the social area, and the number of transitions into the social area.

#### 2.4.7. T-Maze Memory Test

Sixteen zebrafish from each concentration were selected for memory test using a T-maze tank with 20 cm arm length, 10 cm height, and 50 cm width. A total of four tanks were used simultaneously, with one fish placed in each tank. Due to the zebrafish’s reported preference for the color green over red [37], each arm was wrapped in either red or green paper, with the remainder of the arm covered in white paper. Before the T-maze test, zebrafish were required to undergo training. During the training phase of 20 min, the zebrafish were free to explore the T-maze. When the zebrafish initially entered the red (non-preference) chamber, additional brine shrimp were provided, and the zebrafish were kept in this zone. If the zebrafish had never entered the red room, a partition would have restricted it to the red room and rewarded it with a brine shrimp for 10 min. Each 20-min training session occurred twice every day, for a total of four training sessions. The T-maze test was conducted the next day after the four training sessions. The training data of this experiment were shown in the Appendix A. During the 5-min T-maze test, behavioral data were collected for each tested zebrafish without an adaptation time. The swimming time and distance of zebrafish in the red arm, green arm, and white zone were collected, along with their first arrival time in the red arm.

### 2.5. Determining the Neurotransmitters in the Brain

After all behavioral tests were completed, zebrafish were euthanized in ice water, decapitated, and their brains were quickly collected on ice. Four fish brains were combined to produce a single sample, and four samples were analyzed for each concentration group. The fish brains were homogenized at low temperature by adding ultrapure water containing 1.89 % formic acid at a ratio of 1 g: 10 mL, followed by 40 min of centrifugation at 14,000 rpm. The supernatant was then transferred to a fresh tube, followed by addition of acetonitrile containing 1% formic acid at a ratio of 1:4 (*v*/*v*), and centrifuged at 14,000 rpm for five minutes. The resulting supernatant was transferred to a new tube, concentrated, and redissolved to a concentration of 2 mL. Determination of neurotransmitters in prepared brain tissues was performed with reference to previous studies [38]. Using external standard procedures, the produced samples were subsequently tested for cortisol, Gln, GABA, DA, and 5-HT. The method of LC–MS analysis utilized to quantify neurotransmitters in this experiment was detailed in the Appendix A.

### 2.6. Statistical Analysis

All results were expressed as mean ± SEM (standard error of the mean). Normality was assessed using Shapiro–Wilk test. Data conforming to a normal distribution were analyzed by one-way ANOVA, followed by Dunnett’s T3 multiple comparisons test using Graph Pad Prism v9.0. When the parametric ANOVA assumptions failed, a Kruskal–Wallis one-way ANOVA on Ranks with Dunn–Bonferroni post-hoc comparison test was used. Significance levels were set at * *p* < 0.05, ** *p* < 0.01, *** *p* < 0.001, and **** *p* < 0.0001.

## 3. Results

### 3.1. Fentanyl Induced Conditional Place Preference and Reduced Cohesion in the Shoaling Behavior of Zebrafish

The average population percentage of zebrafish in the drug-paired zone (black spots) tended to increase with increasing fentanyl concentration (Figure 2B). Zebrafish in the 100 mg/L and 1000 mg/L concentration groups exhibited higher mean number percentages on the drug-paired side, which was significantly different from the control group (one-way ANOVA, F (3, 36) = 8.646, *p* = 0.0002; Dunnett’s T3 post hoc test, *p* = 0.0003 for 100 mg/L, *p* = 0.0216 for 1000 mg/L). A 10 mg/L concentration of fentanyl did not significantly increase the place preference of zebrafish compared to the control group (Dunnett’s T3 post hoc test, *p* = 0.9568). The nearest adjacent distance (NND) was significantly increased in zebrafish treated with 1000 mg/L fentanyl compared to control fish (Figure 2C, one-way ANOVA, F (3, 76) = 2.644, *p* = 0.0552; Dunnett’s T3 post hoc test, *p* = 0.035). Moreover, the interindividual distance (IID) increased significantly in zebrafish after injection of 100 mg/L and 1000 mg/L fentanyl compared to the control group (Figure 2E, one-way ANOVA, F (3, 76) = 10.66, *p* < 0.0001; Dunnett’s T3 post hoc test, *p* = 0.0054 for 100 mg/L, *p* < 0.0001 for 100 mg/L). No significant alterations in mean speed of zebrafish due to fentanyl treatment compared to control were observed (Figure 2D). These results indicated that fentanyl at 100 mg/L and 1000 mg/L reduced cohesion in the shoaling.

### 3.2. Fentanyl Reduced Individual Zebrafish Anxiety in a Novel Tank Test

The novel tank test proved to be a useful method for anxiety behavior, and increased preference for the top zone in zebrafish was associated with reduced anxiety. We evaluated the distance and duration traveled by individual zebrafish in the top and bottom regions at each concentration for 5 min and found that zebrafish in the 0, 10, 100, and 1000 concentration groups substantially favored the bottom region, exhibiting anxiety-like behaviors (Figure 3A). The movement distance of zebrafish in each fentanyl-treated group was not significantly different compared to the control group (Figure 3B) (one-way ANOVA, F(3, 58) = 2.594, *p* = 0.0611), but zebrafish in the 1000 mg/L fentanyl group significantly increased time spent in the top area (Figure 3C) (Kruskal–Wallis test, H (4, 62) = 12.6, *p* = 0.0056; Dunnett’s T3 post hoc test, *p* = 0.0014) and decreased latency to enter the top area (Figure 3D) (Kruskal–Wallis test, H (4, 62) = 12.79, *p* = 0.0051; Dunnett’s T3 post hoc test, *p* = 0.0015). All fentanyl-treated groups exhibited significantly increased transitions to the top area (Figure 3E) (Kruskal–Wallis test, H (4, 62) = 12.53, *p* = 0.0058; Dunnett’s T3 post hoc test, *p* = 0.0363, *p* = 0.0044, and *p* = 0.0166, respectively).

### 3.3. Fentanyl Reduced Individual Zebrafish Anxiety in Novel Object Exploration Test

Cognition is an important neuro-behavioral trait that can be used to investigate negative consequences of psychoactive substances. By setting up environmentally friendly green new objects, we evaluated the influence of fentanyl on zebrafish’s ability to cognize new items. The movement distance of fentanyl-treated zebrafish tended to decrease with increasing fentanyl concentration during the 5 min of testing, but the difference was not significant compared to the control group (Figure 4B) (one-way ANOVA, F (3, 58) = 1.092, *p* = 0.3600). Zebrafish treated with high doses of fentanyl (1000 mg/L) moved significantly less distance in the area of the new object (Figure 4C) (one-way ANOVA, F (3, 58) = 2.269, *p* = 0.0900; Dunnett’s T3 post hoc test, *p* = 0.0473) and showed a significant reduction in the number of entries into the new object area (Figure 4D) (one-way ANOVA, F (3, 58) = 4.829, *p* = 0.0045; Dunnett’s T3 post hoc test, *p* = 0.0221), indicating that fentanyl caused a decrease in the preference for exploratory behavior. There is no significant difference in latency to enter the new object area with fentanyl exposure compared to the control fish (Figure 4E, Kruskal–Wallis test, H (4, 62) = 1.313, *p* = 0.7260).

### 3.4. Fentanyl Increased Aggressive Behavior of Individual Zebrafish in Mirror Attack Test

Evaluation of the aggressive behavior of adult zebrafish by means of mirror assault is a popular approach. In our study, zebrafish treated with fentanyl showed no significant difference in movement distance compared with the control group (Figure 5B) (Welch’s ANOVA test, F (3.000, 32.07) = 1.713, *p* = 0.1839). The zebrafish in the 100 mg/L concentration group moved significantly faster than the control group (Figure 5C) (one-way ANOVA, F (3, 59) = 2.451, *p* = 0.0723; Dunnett’s T3 post hoc test, *p* = 0.0248). In the evaluation of aggression, all zebrafish in the fentanyl group spent more time in the mirror area than the control fish, but the difference was not significant (Figure 5D) (one-way ANOVA, F (3, 59) = 1.770, *p* = 0.1627). Fentanyl did not cause significant changes in transitions to the mirror area (Figure 5E) (one-way ANOVA, F (3, 59) = 0.1272, *p* = 0.9436). However, zebrafish treated with 100 mg/L and 1000 mg/L fentanyl showed significantly increased duration of fast swimming time in the mirror area compared to the control (Figure 5F) (one-way ANOVA, F (3, 59) = 3.821, *p* = 0.0143; Dunnett’s T3 post hoc test, *p* = 0.0086 for 100 mg/L, *p* = 0.0303 for 1000 mg/L).

### 3.5. Fentanyl Reduced the Social Preferences of Individual Zebrafish in Social Preference Test

Zebrafish have a preference for collective action; thus, we examined the neurotoxicity of fentanyl by observing changes in their social preferences. There were no significant differences in the distance travelled by the fentanyl-treated zebrafish compared to the control (Figure 6B) (one-way ANOVA, F (3, 60) = 3.942, *p* = 0.0124; Dunnett’s T3 post hoc test, all *p* > 0.05). The zebrafish in the 100 mg/L and 1000 mg/L concentration groups spent significantly less time moving in the social zone than control zebrafish (Figure 6C) (one-way ANOVA, F (3, 60) = 5.869, *p* = 0.0014; Dunnett’s T3 post hoc test, *p* = 0.0138 for 100 mg/L, *p* = 0.0133 for 1000 mg/L), but there was no difference in transitions to the social area (Figure 6D) (one-way ANOVA, F (3, 60) = 0.3768, *p* = 0.7700). In addition, the latency for zebrafish to enter the social area after fentanyl treatment had no significant difference (Figure 6E) (Kruskal–Wallis test, H (4, 64) = 2.768, *p* = 0.4288). The results indicated that fentanyl at 100 mg/L and 1000 mg/L reduced the social preference of zebrafish.

### 3.6. Fentanyl Impaired Learning Memory Capacity in the T-Maze Memory Test

Food stimulation was used to train zebrafish on their non-preferred red arm, and color preference changes were used to assess food-induced spatial memory ability. In our study, control zebrafish showed no significant preference for the green arm at the end of training in terms of distance moved (one-way ANOVA, F (2, 45) = 0.2250, *p* = 0.7994) and time spent (one-way ANOVA, F (2, 45) = 0.4487, *p* = 0.6413) compared to the red arm, suggesting that the original green preference has changed due to food induction (Figure 7A–C). However, zebrafish treated with 10 mg/L, 100 mg/L, and 1000 mg/L of fentanyl moved significantly greater distances in the green area than in the red area (Figure 7B) (for 10 mg/L: one-way ANOVA, F (2, 39) = 2.769, *p* = 0.0751, Dunnett’s T3 post hoc test, *p* = 0.0476; for 100 mg/L: one-way ANOVA, F (2, 45) = 5.906, *p* = 0.0053, Dunnett’s T3 post hoc test, *p* = 0.0029; for 1000 mg/L: one-way ANOVA, F (2, 39) = 6.120, *p* = 0.0049, Dunnett’s T3 post hoc test, *p* = 0.0025). In addition, fentanyl-treated zebrafish from 100 mg/L and 1000 mg/L spent significantly more time in the green area than in the red area (Figure 7C) (for 100 mg/L, Kruskal–Wallis test, H (3, 48) = 8.373, *p* = 0.0152, Dunnett’s T3 post hoc test, *p* = 0.0356; for 1000 mg/L, Kruskal–Wallis test, H (3, 42) = 11.79, *p* = 0.0028, Dunnett’s T3 post hoc test, *p* = 0.0069). Importantly, the time to first entry into the red arm was considerably longer in the fentanyl group than in the control group (Figure 7D) (Kruskal–Wallis test, H (4, 56) = 10.67, *p* = 0.0137, Dunnett’s T3 post hoc test, all *p* < 0.05). These findings showed that conditioned place stimulation of food altered zebrafish color preferences in the control group but not in the fentanyl group, implying that fentanyl might impair learning and memory of zebrafish.

### 3.7. Fentanyl Disturbs Neurotransmitter Stability in Zebrafish

After behavioral testing, the levels of Gln, GABA, DA, 5-HT, and cortisol in zebrafish brain were measured by LC–MS/MS. From the results, we found that the Gln, GABA, DA, and 5-HT levels in the zebrafish brains of the 1000 mg/L group were significantly increased compared to the control group (Figure 8A–D). However, there was no significant difference in the levels of cortisol in the brains of zebrafish between the fentanyl groups and the control group (Figure 8E). The disruption of neurotransmitters caused by high concentrations of fentanyl was consistent with the results of the concentration at which behavioral changes occurred.

## 4. Discussion

Zebrafish are an effective vertebrate model for behavioral research [17] to evaluate the rewarding effects of new psychoactive substances via self-administration [39] and conditioned location preference [24]. In this study, the rewarding effects induced by different concentrations of fentanyl were evaluated using enhanced conditioned place preferences in the shoaling behavior of zebrafish, along with an analysis of behavioral changes in individual zebrafish, including anxiety, cognition, aggressive behavior, socialization, and memory. To the best of our knowledge, this is the first study to propose that fentanyl can induce changes in conditioned place preference in the shoaling behavior of zebrafish, which may be useful for development of new models of addiction and related mechanistic studies.

On zebrafish models, conditioned place preference tests have been used to evaluate the rewarding effects of drugs, such as morphine [40], methamphetamine [41], ethanol [42], and nicotine [43], based on visual memory effects during drug stimulation. Studies utilizing zebrafish to explore drug reward effects have typically focused on examining individual responses, perhaps due to the ease with which they can be quantified. For example, 27 psychoactive substances have been demonstrated to produce conditioned place preference in adult zebrafish in studies, suggesting that zebrafish models are as reliable as rodent models in researching reward behavior [24]. According to reports, daily exposure to a solution of fentanyl at a dosage of 0.03 μM (10 μg/L) for 20 min over three days can successfully establish conditioned place preference in adult zebrafish. Pre-screening of individual fish preferences was critical in these trials to avoid fish showing strong place preferences or encountering fish with abnormal activity, which could result in increased experimental costs.

In our experiments, zebrafish shoaling behavior showed a preference for fentanyl-accompanied drug zones in the CPP test. Zebrafish exhibit a unique group behavior known as shoaling, which is becoming an attractive model for studying the effects of drugs on the social behavior of organisms and examining impaired social functioning [44,45]. The differential performance of zebrafish in completing maze tasks suggested that there may be a mechanism in the behavioral pattern of zebrafish in the shoals that reinforces the zebrafish’s place learning task [46]. A place preference test on a population of 10 juvenile fish aged 2 weeks revealed that juvenile zebrafish pre-exposed to 0.8 μM of morphine preferred the morphine-containing compartment in a “choice chamber paradigm”, possibly due to morphine-induced sensitization of neural circuits prior to exposure [47]. Although shoaling behavior was not observed in 2-week-old zebrafish larvae, the findings suggested that the population’s place preference inventory could be used to evaluate the rewarding effects of opioids for testing the impact of a variety of addictive substances on the developing vertebrate brain. Some drugs with abuse risk for humans, such as ketamine, meperidine, pentobarbital, and diazepam, failed to induce CPP in zebrafish when administered by immersion in solution [24]. Therefore, considering the uncertainty of the time and efficiency of drug absorption by immersion in zebrafish, as well as the inability to provide precise data for dose conversion, we preferred intraperitoneal injection in this experiment. Although the concentrations of fentanyl used in our experiments were higher than the blood concentrations reported in fentanyl-related deaths (e.g., 24 µg/L and 57.9 µg/L), no mortality was observed in zebrafish during the experiments, which may be due to interspecies variability. In comparison to previous studies of conditioned place preference in zebrafish induced by intraperitoneal injections of 40 mg/kg METH mg/kg [41] or 40 mg/kg morphine [40], our findings (1000 mg/L, 5 μL, approximately 10 mg/kg) showed that fentanyl could induce rewarding effects in fish at lower doses.

In zebrafish shoaling behavior and social preference trials, fentanyl demonstrated reduced social behavioral effects, which may affect zebrafish populations’ reactions to alarm compounds [48]. We discovered that fentanyl reduced shoal cohesion and individual zebrafish social preferences, which may be related to decreased cortisol levels [49]. In addition, fentanyl-induced higher distances and time spent in the mirror area did not represent a simple interaction with the virtual image but rather a response to social behavior [50]. Overactive aggression in social behavior may be associated with increased GABA levels [51], which has been found in Syrian hamsters (Mesocricetus auratus) treated with low-dose (0.5 mg/kg/day) cocaine [52]. Fentanyl decreased zebrafish social preference and collective cohesion, potentially due to a homeostatic imbalance of glutamate and dopamine neurotransmitters and hormone of cortisol.

High concentrations of fentanyl reduced anxiety levels in individual zebrafish in the new tank dive test (NTDT). The NTDT is the most commonly used anxiety paradigm in zebrafish [53], and the duration spent at the bottom of the test tank and frequency of transitions to the upper compartment are regarded as indications of zebrafish anxiety. Prior research has demonstrated that opioids have a sedative impact on fish [54]. Although not reported in adult zebrafish, an investigation with juvenile zebrafish revealed that exposure to fentanyl impaired juvenile zebrafish’s behavioral ability to respond to light and dark stimuli [21]. We did not find a significant decrease in cortisol levels in zebrafish brains, and anxiety levels are usually correlated with cortisol levels. However, it has also been demonstrated that the sedative effect of fentanyl has a temporary effect as rats given large doses of fentanyl (0.3 mg/kg/d) over extended periods of time demonstrated a reduction in anxiety-like behavior [55]. In contrast, mice exhibited anxious behavior during the fentanyl withdrawal cycle [56]. Consequently, the sedation and anxiety generated by short-term fentanyl administration may fluctuate over time.

The novel object recognition test (NORT), which utilizes the tendency of most vertebrates to explore new things rather than familiar ones [57], is the most commonly used method for assessing recognition memory. Our findings suggested that fentanyl altered cognitive preferences in zebrafish, which was consistent with reports that long-term use of most addictive drugs, including cannabinoids, ketamine, methamphetamine, and cocaine, reduced overall cognitive function [58]. Injection of fentanyl did not induce an effective food-induced conditioned location preference in zebrafish, which may be consistent with the long-term damage caused by anesthetics. Different anesthetics significantly altered hippocampal network dynamics, synaptic connectivity, and consolidation of memory [10]. Fentanyl could also affect the dynamics of the hippocampal network, synaptic connections, and consolidation of memory [59].

Studies have demonstrated that opioid dependence and addiction cause central-nervous-system-homeostasis-related brain effects [60]. Neurotransmitters, as important chemical small molecules, perform the function of neural signal transmission between cells. Drug addiction is associated with disruption in a large number of biomarkers in various areas of the brain, especially in the reward centers. For example, high neurotransmitter concentrations are often closely associated with brain diseases such as Alzheimer’s disease, depression, schizophrenia, and Parkinson’s disease [61]. Neurotransmitter levels in the central reward region of the brain can influence neurobehavior, with dopaminergic and serotonergic neurons associated with drug-seeking behavior in humans [62]. Dopamine is the most important catecholaminergic neurotransmitter in the vertebrate brain, involved in regulation of a range of brain functions related to cognition, such as fear, emotion, motivation, attention, movement, reward, and memory, and its concentration changes are influenced by drug addiction [63,64,65].. Serotonin is one of the main neurotransmitters of the central nervous system (CNS), regulating perception, aggression, and anxiety disorders, sexual behavior, appetite, vascular function, and pain [66,67]. Experiments with fentanyl in rodents suggested that fentanyl induced opioid-triggered extracellular dopamine in the nucleus accumbens shell (NACSh), while elevated aminobutyric acid concentrations may also be involved in opioid reward, which was consistent with our findings in zebrafish [68,69]. Glutamatergic neurons are the most important excitatory neurons in the central nervous system, regulating the amount of glutamatergic transmitters released into the synaptic gap and participating in almost all functional brain activities. L-glutamine (Gln) is the most abundant amino acid in plasma and cerebrospinal fluid and is the precursor of the major central nervous system excitatory (L-glutamate) and inhibitory (gamma-aminobutyric acid) neurotransmitters. Increased glutamine increases formation of GABA, leading to excitotoxicity and impairment in spatial learning and memory [61,70]. In our experiments, major neurotransmitters (including Gln, GABA, DA, and 5-HT) and cortisol exhibited disturbed homeostasis. However, only the 1000 mg/L fentanyl exposure group significantly induced neurotransmitters disturbance; the results indicated that disorders of Gln, GABA, DA, and 5-HT may contribute to the movement disorders induced by 1000 mg/L fentanyl exposure. Meanwhile, there was no significant change in the four neurotransmitters after 100 mg/L fentanyl exposure, suggesting that 100 mg/L fentanyl exposure could not change the transmitters levels, although there was a slight change in behavior. However, drug-withdrawal-induced responses, characterized by increased anxiety and elevated systemic cortisol levels, were not observed in this study [35], which was likely due to the fact that we did not observe them for a sufficient amount of time. Another shortcoming is that we did not further analyze the metabolites of the relevant neurotransmitters, which may provide additional evidence for changes in the relevant neurotransmitters.

## 5. Conclusions

In summary, we investigated fentanyl-induced changes in conditioned place preference in zebrafish shoaling behavior, followed by a series of behavioral tests on individual zebrafish, including novel tank, novel object exploration, mirror attack, social preference, and a T-maze memory test, to determine the potential neuro-toxicological effects of fentanyl. The content of neurotransmitters in the brains of adult zebrafish was also measured. The findings revealed that a high concentration of 1000 mg/L fentanyl induced conditioned place preferences of shoaling behavior, increased aggression, reduced socialization, anxiety, and memory. Disturbances in neurotransmitter levels of Gln, GABA, DA, and 5-HT and hormone level of cortisol in the zebrafish brain may be associated with behavioral changes. These results provided a method for evaluating behavioral alterations and neurotoxicity associated with fentanyl and other drug abuse.

## Figures and Tables

**Figure 1 ijerph-19-13533-f001:**
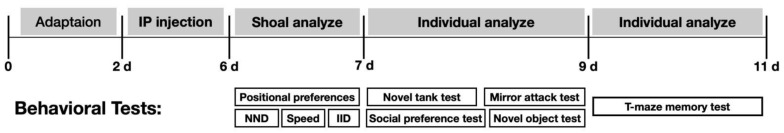
Schematic of zebrafish behavior tests at different time points.

**Figure 2 ijerph-19-13533-f002:**
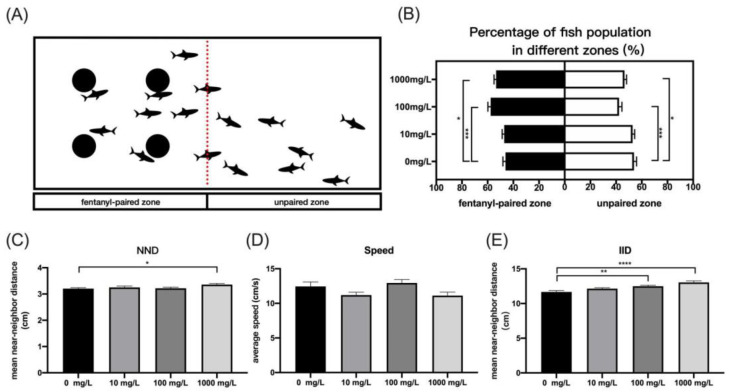
Characteristics of zebrafish shoaling behavior after fentanyl injection. (**A**) Place preferences for fentanyl-accompanied (black spots) and non-accompanied zone in the shoaling behavior of zebrafish. (**B**) The percentage of mean number of zebrafish on the drug-paired zone and the unpaired zone. (**C**) The nearest neighbor distance (NND). (**D**) Average speed. (**E**) The inter-individual distance (IID). Data were expressed as mean ± SEM. All behavioral parameter data were analyzed by one-way ANOVA followed by Dunnett’s T3 multiple comparisons test. The level of significance was defined as * *p* < 0.05, ** *p* < 0.01, *** *p* < 0.001, or **** *p* < 0.0001.

**Figure 3 ijerph-19-13533-f003:**
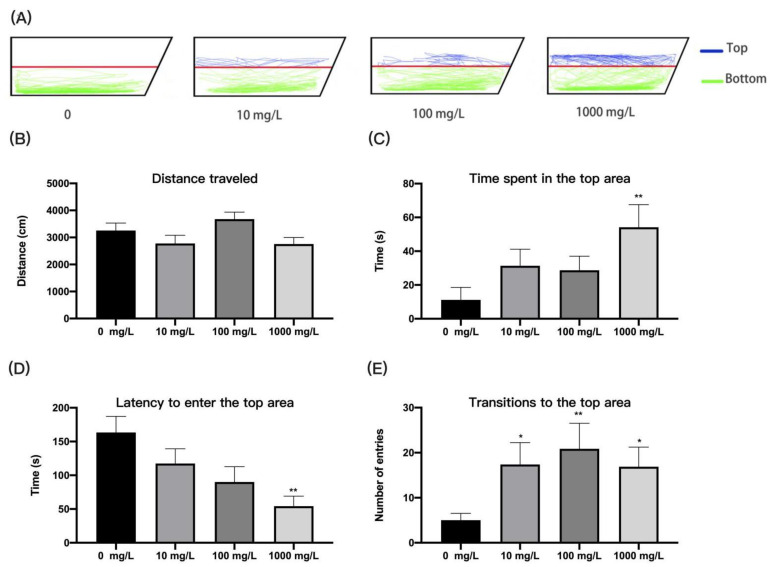
Behavioral characteristics of zebrafish after fentanyl injection in the novel tank test. (**A**) Typical swimming trajectory of zebrafish in each group. (**B**) The distance of zebrafish travelled in the tank. (**C**) The time spent in the top area. (**D**) Latency to enter the top area. (**E**) Transitions to the top area. Data were expressed as mean ± SEM. All behavioral parameter data were analyzed by one-way ANOVA or Kruskal–Wallis test followed by Dunnett’s T3 multiple comparisons test. The level of significance was defined as * *p* < 0.05 or ** *p* < 0.01.

**Figure 4 ijerph-19-13533-f004:**
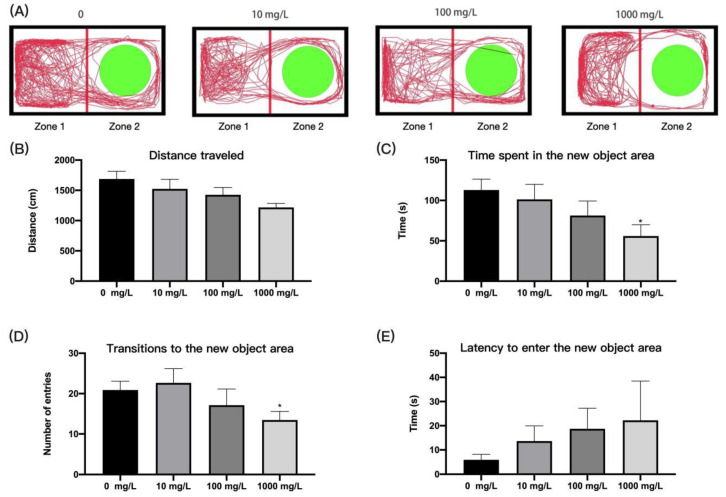
Behavioral characteristics of zebrafish after fentanyl injection in novel object exploration test. (**A**) Typical swimming trajectory of zebrafish in each group. (**B**) The distance of zebrafish travelled in the tank. (**C**) The time spent in the new object area. (**D**) Transitions to the new object area. (**E**) Latency to enter the new object area. Data were expressed as mean ± SEM. All behavioral parameter data were analyzed by one-way ANOVA or Kruskal–Wallis test followed by Dunnett’s T3 multiple comparisons test. The level of significance was defined as * *p* < 0.05.

**Figure 5 ijerph-19-13533-f005:**
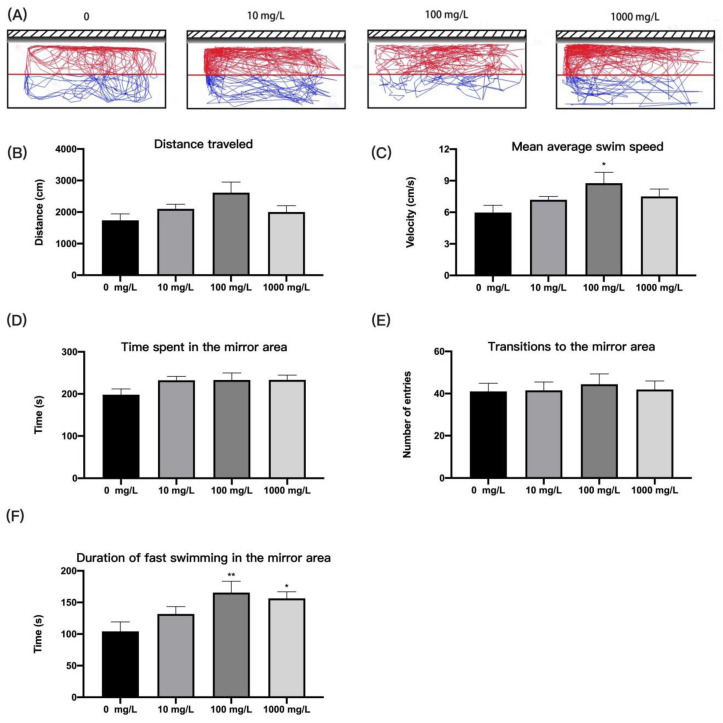
Behavioral characteristics of zebrafish after fentanyl injection in mirror attack test. (**A**) Typical swimming trajectory of zebrafish in each group. (**B**) The distance of zebrafish travelled in the tank. (**C**) The mean average swim speed of each group in the mirror attack test. (**D**) The time spent in the mirror area. (**E**) Transitions to the mirror area. (**F**) Duration of fast swimming in the mirror area. Data were expressed as mean ± SEM. All behavioral parameter data were analyzed by one-way ANOVA followed by Dunnett’s T3 multiple comparisons test. The level of significance was defined as * *p* < 0.05, ** *p* < 0.01.

**Figure 6 ijerph-19-13533-f006:**
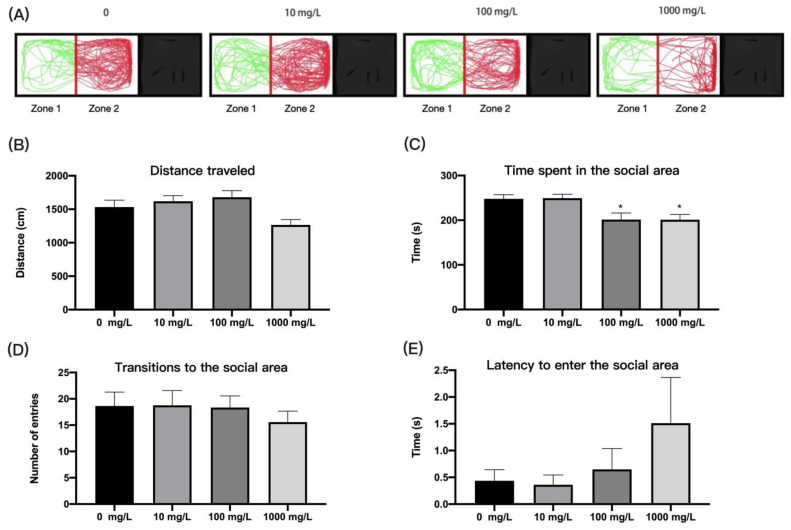
Behavioral characteristics of zebrafish after fentanyl injection in social preference test. (**A**) Typical swimming trajectory of zebrafish in each group. (**B**) The distance of zebrafish travelled in the tank. (**C**) The time spent in the social area. (**D**) Transitions to the social area. (**E**) Latency to enter the social area. Data were expressed as mean ± SEM. All behavioral parameter data were analyzed by one-way ANOVA or Kruskal–Wallis test followed by Dunnett’s T3 multiple comparisons test. The level of significance was defined as * *p* < 0.05.

**Figure 7 ijerph-19-13533-f007:**
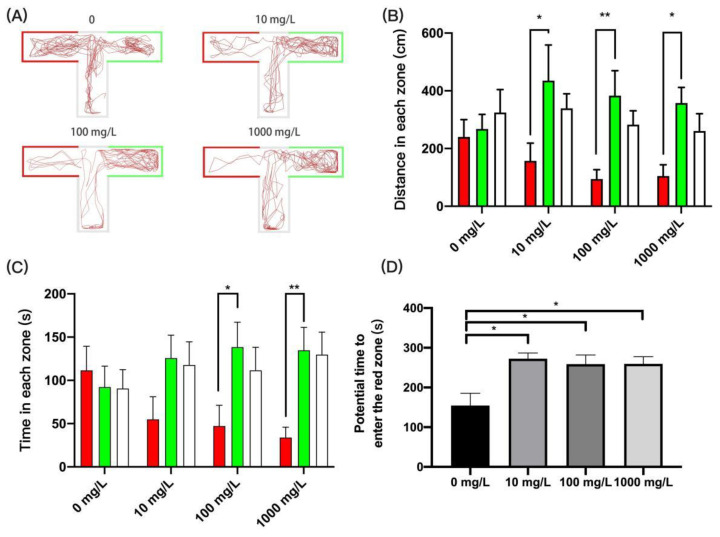
Behavioral characteristics of zebrafish after fentanyl injection in T-maze test. (**A**) Typical swimming trajectory of zebrafish in each group. The red arm is the food reward area; the green arm is the non-food reward area. (**B**) The distance of zebrafish travelled in red zone, green zone, and white zone. (**C**) The time of zebrafish travelled in red zone, green zone, and white zone. (**D**) The time to first entry into the red zone. Data were expressed as mean ± SEM. All behavioral parameter data were analyzed by one-way ANOVA followed by Dunnett’s T3 multiple comparisons test. The level of significance was defined as * *p* < 0.05 or ** *p* < 0.01.

**Figure 8 ijerph-19-13533-f008:**
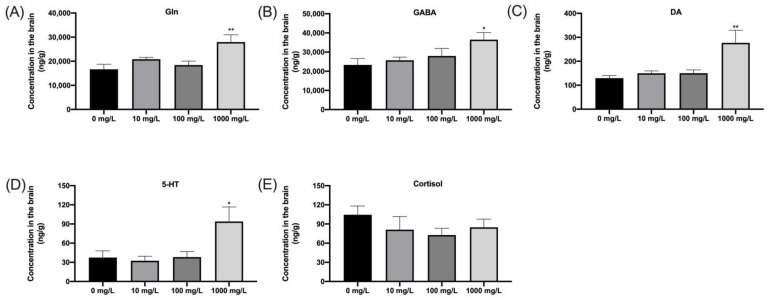
Effects of fentanyl administration on brain glutamine (Gln), γ-aminobutyric acid (GABA), dopamine hydrochloride (DA), 5-hydroxytryptamine (5-HT), and cortisol, assessed by high-performance liquid chromatography (HPLC). (**A**) Gln. (**B**) GABA. (**C**) DA. (**D**) 5-HT. (**E**) Cortisol. Data were expressed as mean ± SEM and analyzed by one-way ANOVA followed by Dunnett’s T3 multiple comparisons test. The level of significance was defined as * *p* < 0.05 or ** *p* < 0.05.

## Data Availability

Not applicable.

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
