# Peer review of "Fentanyl Induces Novel Conditioned Place Preference in Adult Zebrafish, Disrupts Neurotransmitter Homeostasis, and Triggers Behavioral Changes"

_ijerph, 2022, doi:10.3390/ijerph192013533_

Round 1
Reviewer 1 Report
This manuscript identified the neurotoxicity of fentanyl in zebrafish through multiple neurobehavioral tests, and they found the fentanyl-induced neurotransmitters disturbance may be due to the behavioral deficits they found in the treated zebrafish. Overall, the experiments they conducted in this study are solid. However, there are several concerns that need to be addressed before this manuscript can be accepted for publication.
1. In the Novel tank test, the authors should provide details of how they define the top and bottom areas.
2. For the T-maze test, when was it conducted? Right after the 4-day training or the next day after the 4 days training?
3. Did the Fentanyl exposure affect the body weights of the fish?
4. The title of result 3.3 is wrong.
5. Please provide the mean average swim speed of each group in the mirror attack test. Also, provide how did the author define the top area.
6. It would help the readers to understand the whole experiment if the authors provide the training data in the T-maze test.
7. The authors found neurotoxicity of 100 mg/L Fentanyl in the behavioral test, however, the major neurotransmitter in the 100 mg/L group was similar to the control. If it is true that the disturbance of these neurotransmitters can induce behavioral deficits, why it is not found in the 100 mg/L group? The authors should discuss it.
Reviewer 2 Report
The present manuscript investigates if zebrafish can be used to test the addictive/neurotoxic potential of psychoactive substances, using fentanyl, a synthetic opioid commonly used in clinical practice, as a study case. The authors conjugate a series of interesting behavioural tests (novel tank, novel object exploration, mirror attack, social preference, and T-maze memory), with data from neurotransmitters and cortisol.
Based on the obtained results, they claim that to have identified a method for assessing the risk of addiction to new psychoactive substances.
This is a clear, well written, nicely illustrated study.
A few issues, however, still require further attention:
1) In the introduction section the authors state that “Up to our knowledge, the shoaling behaviour of zebrafish has not been used in CPP tests to investigate the drug reward effects”. However, they should also refer at this point that CPP has been used in this context already, as is the following papers:
.
Ponzoni L, Braida D, Bondiolotti G, Sala M. The Non-Peptide Arginine-Vasopressin v1a Selective Receptor Antagonist, SR49059, Blocks the Rewarding, Prosocial, and Anxiolytic Effects of 3,4-Methylenedioxymethamphetamine and Its Derivatives in Zebra Fish. Front Psychiatry. 2017 Aug 14;8:146. doi: 10.3389/fpsyt.2017.00146. PMID: 28855876;
Braida D, Limonta V, Pegorini S, Zani A, Guerini-Rocco C, Gori E, et al. Hallucinatory and rewarding effect of salvinorin A in zebrafish: kappa-opioid and CB1-cannabinoid receptor involvement. Psychopharmacology (Berlin) (2007) 190:441–8. 10.1007/s00213-006-0639-1
Stewart A, Riehl R, Wong K, Green J, Cosgrove J, Vollmer K, et al. Behavioral effects of MDMA (’ecstasy’) on adult zebrafish. Behav Pharmacol (2011) 22:275–80. 10.1097/FBP.0b013e328345f758
2) Methods, section 2.5, please clarify the meaning of “sedated” for animals placed on ice.
3) Legend o figure 7: Please clarify also in the legend the meaning of “green” and “red” areas.
4) Reference 38 is missing from the references list.
5) Discussion, L483: The trend toward reduced cortisol is not clear and should not be used to justify behavioural data, please adjust this section in accordance.
6) The last section of Discussion refers to neurotransmitters. Several sentences used here are too telegraphic, please rewrite towards a more explanatory text. Also, these data fails to measure the metabolites of dopamine and serotonin, which make the interpretation of such results more difficult. This issue should be discussed.
Reviewer 3 Report
The manuscript «Fentanyl induces novel conditioned….” by Wu et al. is potentially interesting and shows impacts on neurotransmitters/substrates and bevavioral tests in zebrafish. It also seem to show that the zebrafish is a good model for this kind of experiments. However, the authors make too many erroneous statements to be accepted in this format. The following should be amended or improved:
- The concentration of fentanyl in studies of fentanyl-related deaths is much lower than the concentrations they use in this study (e.g., 24 ug/L and 57.9 ug/L versus 1000mg/L). This may give additional toxic effects. This should be discussed properly.
- Unclear the difference between “The nearest neighbor distance” and “the inter-individual distance”. This should be explained.
- The authors write “The behavioral ability of zebrafish …..” and refer to Figure 3B. However, on the figures it is written that “Distance traveled”. Is behavioral ability the same as distance traveled? This should be more clear.
- The title of 3.3 is the same as 3.2. This has to be changed.
- On many occasions they write that something has changed, but it is not significant. E.g. on page 10: “…Latency for zebrafish…..showed an increasing trend with concentration, but the differencewas not significant…” The changes are not significant, therefore you can not report a change or a trend here. If you do it, you must also say that the trend is reduction from 0 mg/L to 10 mg/L before it is increasing. Similar sentences you find on page 8, 9, 12 and other places. They should be amended.
- fig. 7: you write “white zone” but it is black on the figure. Amend.
- Gutamine is abbreviated to Gln and glutamate to Glu. You cannot use Glu for glutamine.
- You report upregulation of glutamine, GABA, DA and 5-HT. And you write on page 14 that “The glutamate –glutamine cycle may be disrupted in rodents due to excessive activation of NMDA receptors, leading to excitotoxitiy and impairment of spatial learning and memore”. But this is wrong. The glutamate-glutamine cycle is about recycling of neuronal glutamate and GABA through astroglial glutamine and then reformation og glutamine. So, NMDA receptors have nothing to do with it. Your data may support that increased glutamine increases formation of GABA, and that this is responsible for the effects you see. See e.g. Qureshi et al., Cells, 2020, Qureshi et al., Cereb Cortex, 2019 on the glutamate-GABA-glutamine cycle and the formation of GABA.
- Cortisol is a hormone, not a neurotransmitter! This must be corrected.
- It would be better to have the results and the accompanying figure on the same page, at least for many of them by placing the figures in a different way.
- Many references are written in the text but not included in the bibliography. E.g. reference 40 and 43.
